# Consumer Awareness, Attitudes and Preferences Towards Heritage Cereals

**DOI:** 10.3390/foods9060742

**Published:** 2020-06-04

**Authors:** Karin Wendin, Arwa Mustafa, Tove Ortman, Karin Gerhardt

**Affiliations:** 1Department of Food and Meal Science, Kristianstad University, 29188 Kristianstad, Sweden; arwa.mustafa@hkr.se; 2Department of Food Science, University of Copenhagen, Rolighedsvej 26, 1958 Frederiksberg, Denmark; 3Department of Urban and Rural Development, Swedish University of Agricultural Sciences, 75007 Uppsala, Sweden; tove.ortman@slu.se (T.O.); karin.gerhardt@slu.se (K.G.)

**Keywords:** heritage cereals, consumer attitudes, preferences and awareness

## Abstract

Interest in heritage cereals is increasing among consumers, bakeries and farmers, and the trends point towards the local production of crops and connect to sustainability. The most known variety is spelt, which has opened up for old landraces such as Oland wheat. Heritage cereals have shown a higher resilience than modern varieties and have the potential to supply the market with alternative products that have an attractive cultural background. Delicious and nutritious products based on heritages cereals have a growing market potential. Consumers’ attitudes and preferences to different products are affected by factors such as age, gender and education. The aim of this study was to investigate and analyse different consumer groups’ awareness, attitudes and preferences toward heritage cereals. The number of respondents who participated in this study and answered the web-based questionnaire was 434. It can be concluded that most consumers are aware of heritage cereals. Geographic background had an influence, while academic background did not. Bread and pasta are the most consumed products and are regarded as the most popular future products to be based on heritage cereals. The most essential factors in bread are taste and flavour, followed by freshness and texture. The origin of the cereal and its health aspects are important; women are more concerned about the origin than men, while older consumers are more concerned about health. Older consumers are also more willing to pay extra for heritage cereal than younger consumers.

## 1. Introduction

Today’s consumer trends are moving more towards the local and regional production of crops (e.g., ancient or heritage crops), mainly due to a rising interest in sustainability [1]. It has been shown that the taste experience of a product is of the greatest importance to the consumers. Furthermore, product claims, such as ancient, natural, organic, or local, are the most likely to have a positive impact on the consumer’s preference and/or choice [2,3,4,5]. High acceptability has, for example, been shown for breads containing Kamut or spelt [6].

Despite the numerous genetic and historical data on the origins of agricultural products, there is no universal definition for modern and older cereals [7]. Ancient cereals, according to Giambanelli et al. (2013) [8], are represented by populations of primitive cereals, which were not subjected to any modern breeding or selection processes (e.g., emmer, einkorn and spelt). What are today named as landraces were originated by farmers using natural selection, consequently saving various seed types year after year [9,10,11]. For convenience, in this paper the term “heritage cereals” is meant to include ancient cereals, landraces and older varieties.

The pursuit of higher yields and the industrialisation of agriculture over the past 150 years, meant that heritage cereals were lost from many parts of the world [11,12,13]. For future sustainability, there is a need to build up resilient agricultural systems [13,14]. Heritage cereals have shown more resilience to drought or other extreme weather circumstances than the modern varieties, which in turn might contribute to a robust agricultural system [15,16]. 

There is currently a trend of revived interest in heritage cereals from consumers, artisan bakeries and farmers [6]. Farmers of organic crops are interested in certain agronomic traits in heritage cereals, which makes them suitable for organic production [17,18,19]. Additionally, the fact that they are often sold at a premium price makes the old varieties highly attractive for farmers [1,20]. Heritage cereals might as well supply the market with new types of products that have an attractive cultural background and connection to authentic stories. Storytelling is highly important for heritage cereals and their growers and is an influential “trademark” [21]. Moreover, the demand for locally produced food is increasing [22,23], and alternative types of distributional and sale systems have gained ground, i.e., “short food supply chains”, in which the heritage cereals fit well. These short supply chains aim to redefine the producer–consumer relationship in terms of providing knowledge of the origin of the food [24]. In the case of Sweden, several initiatives promoting a direct contact between producer and consumer have emerged, with examples such as “Farmer’s market”, “Local Food Nodes” and “REKO-rings”. 

Encouraging the production and consumption of heritage cereals is in line with the Swedish food strategy and the current government goal to increase organic food production [25]. Cereal-based food products constitute a large and central part of the human diet and ancient cereals are suggested to possess health-promoting effects due to their unique nutritional content. Thus, the development of delicious products based on these ancient cereals may enhance the large market potential as well as boost the consumption of whole grain [1,26,27]. 

Consumers’ attitudes and preferences for different kinds of products may differ according to factors such age, gender, education level and geographic background. For example, in the case of fruits and berries, it has been shown that sustainability aspects are of higher importance to women—mainly to younger women—than to men. In the case of bread consumption, Sandvik et al. [2] pointed towards a more traditional consumption structure among Swedish consumers, however, a lower consumption of rye and whole-grain bread could be observed among younger consumers. This is in accordance with other studies showing that older consumer groups are more concerned with health aspects in comparison to younger consumer groups [28]. Consumers with a higher educational level are more aware of the health aspects and are more receptive to trends [29,30], however, knowledge about the level of impact from education is low. Geographical and cultural backgrounds are further factors that might have an impact on the consumers. Thus, it is of interest to study awareness, knowledge and attitudes towards heritage cereals among different groups of consumers in higher education arenas.

## 2. Aim

The aim of this study was to investigate and analyse consumers’ awareness, attitudes and preferences towards heritage cereals. A further aim was to study whether consumers differing in academic and geographic backgrounds varied in the mentioned aspects while taking age and gender into account.

## 3. Materials and Methods

### 3.1. Consumers

Swedish consumers from two different academic institutes in Sweden were invited to answer a questionnaire concerning awareness, attitudes and preferences towards heritage cereals. The academic locations were the Swedish University of Agricultural Sciences (SLU), which is a university with disciplines focusing on primary agricultural production, and Kristianstad University (HKR), which is a university with a multidisciplinary focus. The participants had to be affiliated with one of the universities, either as a student or as an employee. Participation was anonymous and voluntarily. To gain enough data for reliable statistical calculations, a minimum number of 100 adult consumers from each academic location aged 18 years or older was aimed for during the recruitment process [31]. 

### 3.2. Questionnaire

The web-based questionnaire was launched during the month of April 2019. The software, Eye Question (version 3.9.7, Logic 8, Elst, The Netherlands) was used for the data collection. The survey contained the following areas of investigation: (a) consumers’ awareness and consumption of heritage cereals; (b) consumers’ attitudes towards heritage cereals; (c) consumers’ preferences of future products with heritage cereals. The different areas of investigation are given in Table 1. The full questionnaire is provided in Appendix A.

### 3.3. Statistical Evaluation

The collected questionnaire data were processed using descriptive and analytical statistics. Mean values and standard deviations were calculated. A multiple comparison test was performed by one-way ANOVAs in conjunction with Tukey’s Post-Hoc Tests to compare groups of consumers. For observed frequency data, a chi-squared test was performed to determine the level of significant differences between the expected frequencies and the observed frequencies. For all statistical calculations, the significance level was set to *p* < 0.05. SPSS (Version 23, IBM, New York, NY, USA) was used throughout the calculations. The free software, Wordle (wordle.net, IBM Corporation, New York, NY, USA) was used to generate a word cloud out of the words used to illustrate which type of bread the study group consumed.

## 4. Results

### 4.1. Consumers

The total number of participating consumers in the questionnaire was 434, of which 311 were women, 117 men and 6 X (unidentified). Details about the participants are shown in Table 2. From the total study population, 323 participants were affiliated with SLU and 111 participants were affiliated with HKR. The SLU participants consisted of 227 women, 92 men and 4 X. Age group 1 consisted of 120 participants, age group 2 consisted of 108 while age group 3 consisted of 95. The HKR participants consisted of 84 women, 25 men and 2 X. Age group 1 consisted of 37 participants, age group 2 consisted of 42 and age group 3 consisted of 32. Since there was a low number of participants in gender group X, the resulting data from this group have not been taken into consideration.

### 4.2. Consumers’ Awareness and Consumption of Heritage Cereals

To get insight about consumers’ awareness concerning heritage cereal varieties, they were presented with different varieties of heritage cereals and asked to identify those that they were familiar with. Figure 1 presents the different varieties of heritage cereals and the frequencies of awareness within the different population sectors. Spelt was the most known variety among the different population sectors, while Halland wheat was the least known. No significant difference between the groups was shown in the awareness of spelt, while chi-squared tests showed that Halland wheat was significantly more (χ^2^ = 5.98; *p* < 0.05) known to HKR than to SLU participants. Additionally, Oland wheat was significantly more (χ^2^ = 9.47; *p* < 0.05) identified by the HKR participants, yet, it was the least known within age group 3. Furthermore, einkorn was significantly more (χ^2^ = 5.97; *p* < 0.05) known to age group 1. Regarding the identification of varieties Kamut (χ^2^ = 9.23; *p* < 0.05) and Halland wheat (χ^2^ = 8.18; *p* < 0.05) there was significant difference in their recognition among the age groups, where they were the least known for age group 3.

The popularity of the consumption of cereal-based products was investigated by the rate of consumption of the products on weekly bases, as shown in Figure 2.

The responses to the question about the cereal products consumed at least once per week revealed that bread, followed by pasta, was the most consumed product among all categories of cereal-based products. Chi-squared tests revealed that the consumption of pasta was significantly lower in age group 3 compared to the younger age groups (χ^2^ = 8.30; *p* < 0.05). Flakes and Muesli were significantly more consumed by men (χ^2^ = 6.66; *p* < 0.05) among SLU participants (χ^2^ = 4.56; *p* < 0.05) and within age group 3 (χ^2^ = 6.30; *p* < 0.05). Figure 2 shows that more than 90% of the study population consumed bread. Figure 3 gives an indication of the most common types of bread consumed among the study population. Among the most widely consumed were crispbread, sourdough bread, sourdough dark bread, rye bread, whole grain and home-baked bread.

### 4.3. Consumers’ Attitudes towards Heritage Cereals

Responses about the consumers’ habits and attitudes when it comes to home baking compared to purchasing sites are illustrated in Figure 4, showing that purchasing at grocery stores was, according to one-way ANOVA, a significantly more common habit among the study population than home-baking or purchasing bread in a baker’s store. There was no significance difference when comparing home-baking or purchasing at a bakery shop, nor was there significant different due to gender or academic institution. However, it was significantly more common to purchase bread at bakery shops in age groups 2 and 3 compared to age group 1 (*f* = 11.30; *p* < 0.05). 

Factors that govern consumers’ preferences for products based on heritage cereals are presented in Table 3.

From Table 3 it can be inferred that Taste/Flavour and Freshness were the most important quality aspects of the bread. When comparing women to men, it was found out by one-way ANOVA that that texture (*f* = 8.82; *p* < 0.05), having the bread made with wholemeal flour (*f* = 14.54; *p* < 0.05) as well as the origin (*f* = 19.64; *p* < 0.05) of the cereal, were significantly more important factors to women than to men. On the other hand, the brand of the bread seemed to be the factor that was least thought about; however, age group 3 seemed to more concerned with this than the younger consumer groups.

When comparing the two institutions, it was found that bread features such as. its appearance (*f* = 4.32; *p* < 0.05), made by sourdough (*f* = 7.61; *p* < 0.05) and its freshness (*f* = 7.83; *p* < 0.05) were significantly more important to the participants from HKR compared to those from SLU.

When investigating bread attributes in relation to age, it was revealed that Odour/Aroma is significantly less important for age group 1 than for older age groups (*f* = 18.65; *p* < 0.05). Sourdough is significantly less important for age group 1 than for age group 3 (*f* = 5.58; *p* < 0.05). Wholemeal is significantly less importance for age group 2 than for age group 1 and age group 3 (*f* = 4.39; *p* < 0.05). The importance of the health aspects of the bread was significantly different among the groups, where it was the most important factor for age group 3 (*f* = 9.41; *p* < 0.05). Being an organic cereal was of significantly less importance for age group 1 than for age groups 2 and 3 (*f* = 12.94; *p* < 0.05). The importance of the price of the bread differed among the groups. Nevertheless, it was significantly of the most important to age group 1 and of the least importance to age group 3 (*f* = 17.27; *p* < 0.05). Freshness of the bread was significantly less important for age group 1 than for age group 2 and age group 3 (*f* = 12.13; *p* < 0.05). 

In Table 3, the mean values and standard deviations are given as well as indications of significant differences.

On the question: “May you consider purchasing bread or other products that are based on heritage cereals?”, as many as 98.4% of the study population responded to consider purchasing bread products based on heritage cereals.

### 4.4. Consumers’ Preferences of Future Products with Heritage Cereals

To explore the future willingness of the study population to purchase cereal products based on heritage cereals, they were presented with set of product categories and were asked which product or products they would consider purchasing. The categories are presented in Figure 5. When comparing gender, the chi-squared test showed that the willingness to purchase porridge was significantly higher among women than men (χ^2^ = 7.45; *p* < 0.05). When comparing age groups, it was revealed that age group 3 was more likely to purchase bread and significantly less probable to purchase pasta when compared to younger age groups (χ^2^ = 21.16; *p* < 0.05). A similar pattern was seen for the purchase of porridge (χ^2^ = 10.47; *p* < 0.05), cooking cereals (χ^2^ = 14.86; *p* < 0.05) and cookies (χ^2^ = 9.57; *p* < 0.05). On the other hand, participants in age group 2 were significantly more likely to purchase flakes (breakfast cereals) (χ^2^ = 7.73; *p* < 0.05) and flour (χ^2^ = 9.56; *p* < 0.05). Bread was the most popular product to be considered purchasing. No differences were observed between SLU and HKR participants.

To be able to get a deeper understanding of the population preferences regarding the accessibility of heritage cereal products, the participants were asked about the location where they would prefer to purchase products. The set of locations are presented in Figure 6. Grocery stores were the most popular site for purchasing, followed by the bakery. However, according to chi-squared analysis, there was no significant difference among group categories. Only some respondents chose the category “others”.

The willingness to pay more for products based on heritage cereals was a common attitude in the different groups of the study population, being more pronounced in age group 3. The chi-squared test showed that age group 1 was significantly the least willing to pay more for products based on heritage cereals in comparison to the other groups (χ^2^ = 9.89; *p* < 0.05), as shown in Figure 7.

## 5. Discussion

This study shows a great consumer interest in heritage cereals, where almost all consumers would consider purchasing bread or other products based on heritage cereals. This may be explained by the health trends and their relation to heritage cereals [32]. Furthermore, this great interest is well supported by respondents’ abilities to identify different varieties of heritage cereals (e.g., more than 95% were aware of the variety spelt). This predominance could, to some extent, be explained by the fact that spelt has a very long history and was used as staple cereal thousands of years ago [33,34,35], and it has been shown that the acceptance of spelt is high among consumers [6]. Over the last few decades, spelt has become more commonly used in baking, and the addition of spelt flour during bread-making gives unique sensory characteristics to the bread (e.g., makes the bread stiffer as well as giving it a prolonged shelf life) [36]. About 40%–50% of the respondents were familiar with other varieties (e.g., emmer, Kamut and Oland wheat). The high percentage of awareness amongst the respondents could probably be explained by their academic background and that many of them belonged to agricultural and food studies departments. Robinson [32] explained European that consumer interest and awareness is influenced by mainstream media, which consequently has become the driving demand for flours from heritage cereals. Swedish consumers, however, are more likely to be influenced by social media and influencers such as Adam Arnesson (@ekobonden), Sebastien Boduet (@sebastienboduet) and many more. It is inferred that respondents from the academic institute HKR had a higher awareness of Oland wheat and Halland wheat compared to SLU, which could be related to the geographic location of the academic institute, where HKR is situated closer to areas where these varieties are cultivated.

It is noteworthy that bread and pasta are the most consumed cereal products. These are also the products that the respondents indicated as most suitable for future heritage cereal products and which they were most willing to purchase. The phenomenal product recognition, in this case bread and pasta, is well known, and for food innovations a combination of recognition, quality, tradition and social approval are very important factors for consumer acceptance [37]. This could also explain the oldest group’s lower interest in pasta. 

This study showed that the most important factors for bread are taste and flavour. This is supported by rising consumer interest for better and more authentic flavours [32,38]. Freshness and texture are other important factors and, according to the respondents, they are more important than other factors, such as health factors, being organic and origin. This is in line with other studies that have reported the importance of flavour and other sensory attributes [28,39]. It was also established that health factors are more important to older consumers than to younger ones [29], an observation that is supported by the results from this study where the oldest consumer group regard health aspects as significantly more important than the younger group.

It is reported that Sweden has a fairly high consumption of organic products [40,41], thus it was surprising that the current study has signalled that the younger respondents regarded a cereal being labelled “organic” less important than the older consumer groups. “Locally produced” has recently been shown as more important to the consumers than “organic farming” [22], suggesting that organic farming would require more land than conventional farming and, in that respect, contribute more to climate change [42].

The word cloud illustration in Figure 3 points out wholegrain as a popular type of bread, which is in line with Kyrø, et al. (2012) [26] who reported on the consumption of bread in the Scandinavian countries during the 1990s, showing that rye contributed the most to the whole-grain intake: in Denmark about 70%; in Sweden about 50%; in Norway only about 20%. Furthermore, the total whole-grain consumption among different Swedish consumer groups were as follows: white-bread consumers had a mean total intake of 38 g/day; whole-grain bread consumers reported 45 g/day [2]. This supports the current study findings that the participants reported crispbread as the most consumed type of bread. Likewise, sourdough bread was reported to be commonly consumed; more common than white-bread and toast-bread. It should be highlighted that the consumers in the current study were affiliated with universities and, therefore, might have had a higher awareness about the role of whole-grain and sourdough for human health. The potential of sourdough to obtain healthier cereal products is becoming increasingly known [43].

In the current study, and based on the above discussion, it is evident that age is a critical factor. For instance, younger consumers are more aware of heritage cereals and different varieties than older consumers. This high-level of awareness is reflected by younger consumers showing a greater interest in natural agricultural products [44]. The younger group was also more sensitive to price and significantly fewer young respondents were willing to pay more for heritage cereals compared to respondents in the older consumer groups, which could be explained by differences in economical levels. Similarly, Hwang [45] showed that older consumers are more willing to pay a higher price for food when they are motivated to do so. This is in line with the results in this study where the older consumers were willing to pay a higher price for products based on heritage cereals than younger consumers. The same pattern could be seen in the habit of purchasing bread at the bakery, which was more frequent among the older consumers.

Another important factor for the older consumer group was the health aspects. Kraus et al. [30] found that food health aspects are of greatest importance for women and older consumer groups in studies on functional food. Gender difference was more obvious when studying the importance of consuming wholegrain products and knowledge about the origin of the cereal. In these assessments it was found out that women considered those as more important factors compared to men. This is also in line with Kraus et al., whose studies show that women are more concerned about nutritional aspects and consuming natural products than men. Moreover, women consumed less muesli and breakfast cereals than men, yet ate more porridge compared to men. 

Regarding the groups differing in academic background, it could be seen that the sites’ geographies seemed to influence the awareness of varieties of heritage cereals. The site of the academic institute seemed to have an impact, mainly on the awareness of different types of heritage cereals, which correlates with a study showing a high awareness of the importance of regional products among consumers [46]. Further, the results indicate that respondents affiliated with SLU consumed more muesli and flakes compared to HKR affiliates, while for respondents from HKR the factors appearance, sourdough and freshness were of higher importance than for those from SLU. These differences and the differences in awareness between academic sites could be due to the fact that HKR has a multidisciplinary focus with research and teaching within many subjects. Therefore, it is possibly more open to influences from a broader number of different disciplines compared to the agricultural focus at SLU.

Limitations: It should be noted that a limitation of this study was the unbalanced sample sizes of the consumer groups. The gender group X (unidentified sex) consisted of only six consumers, thus this group was too small to imply any relevant results and was, therefore, kept out of analysis. It should as well be noted there was an uneven sample sizes regarding men and women, as well as consumers belonging to the different academic institutes. To compare the frequencies of groups, the percentages of frequencies were calculated. It should be noted that a larger sample size of women is common in consumer studies [39,47,48]. Further, it should be noted that the two participating universities differ in size, where SLU is substantially larger than HKR.

## 6. Conclusions

It could be concluded that most consumers are aware of heritage cereals, where dinkle/spelt is the most well-known variety. Other varieties such as emmer, Kamut and Oland wheat were known by approximately 50% of the consumers. The geographic location of the academic institutions seemed to influence the awareness of heritage cereal varieties. The focus on academic background seemed to have only minor influence on attitudes towards heritage cereals. Bread and pasta are the most consumed products and are also regarded as the most potential future products that could be based on heritage cereals. With regards to bread, the most important factor is taste and flavour, followed by freshness and texture. Cereal origin and health aspects are of importance, however, women are significantly more concerned about the origin of the cereal than men, while older consumers are more concerned about health aspects of cereals and cereal-based products. Older consumers are also significantly more willing to pay more for products based on heritage cereal than younger consumer groups.

## Figures and Tables

**Figure 1 foods-09-00742-f001:**
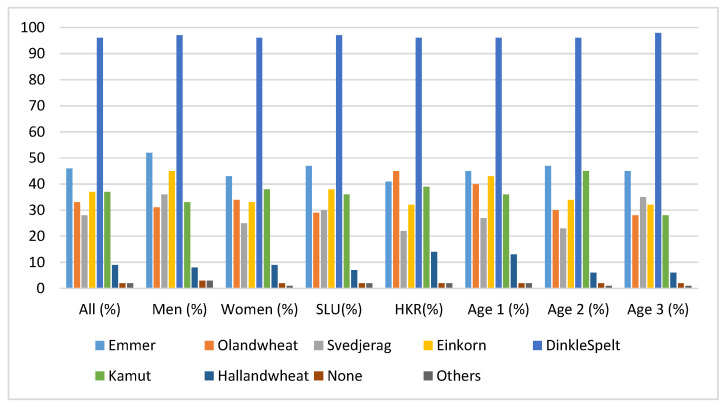
Frequency data given in percentages for each group showing the awareness of the different varieties of heritage cereals. The category “others” included black oat, quinoa, buck wheat, dala wheat, spring wheat, millet, naked oat, and teff. Swedish University of Agricultural Sciences (SLU), Kristianstad University, Sweden (HKR).

**Figure 2 foods-09-00742-f002:**
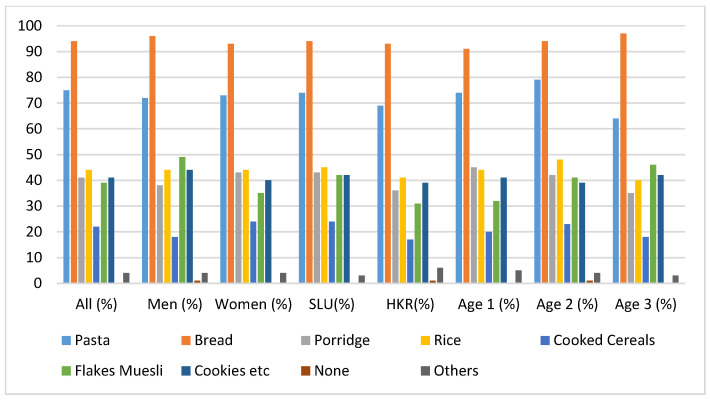
The frequency of consumption given in percentages for each group of cereal based products per week. The category “others” included couscous, rice cookies, crispbread, beer, pancakes, quinoa, millet, seeds, gluten free, buck wheat, and bulgur.

**Figure 3 foods-09-00742-f003:**
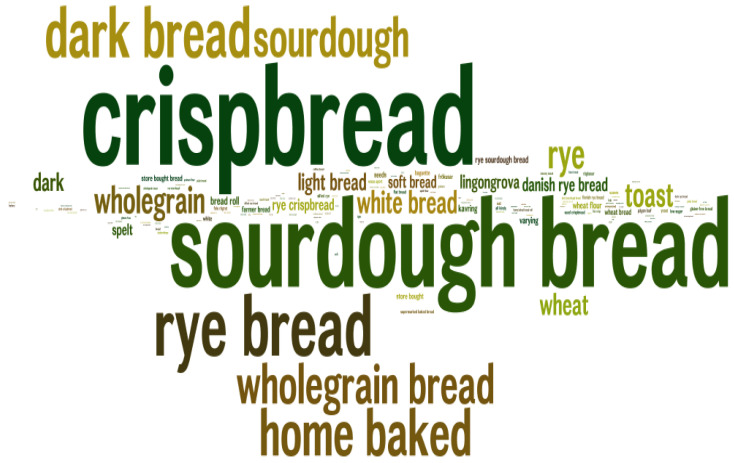
Illustration giving an indication of the popularity of bread types consumed. The figure is based on the qualitative data given in the questionnaire.

**Figure 4 foods-09-00742-f004:**
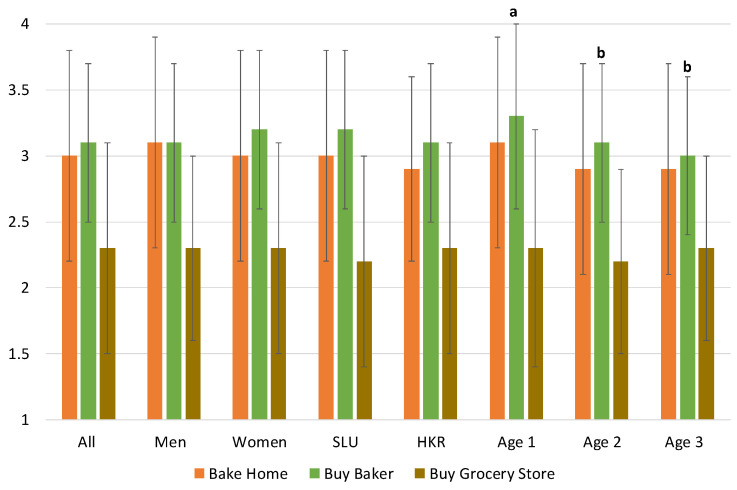
Mean and standard deviations showing trends of purchasing at different sites. Significant differences are indicated with different letters. The scale was 1–4, where 1 = always, and 4 = never.

**Figure 5 foods-09-00742-f005:**
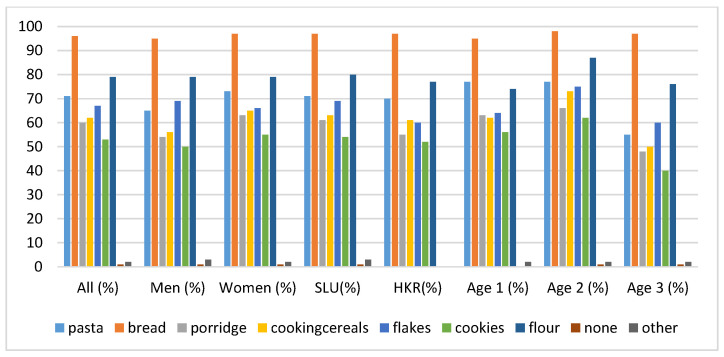
Future willingness to purchase heritage cereal products. The category “others” includes drinks, smoothies, beer brewing, alcoholic beverages and everything today that is done by modern cereals.

**Figure 6 foods-09-00742-f006:**
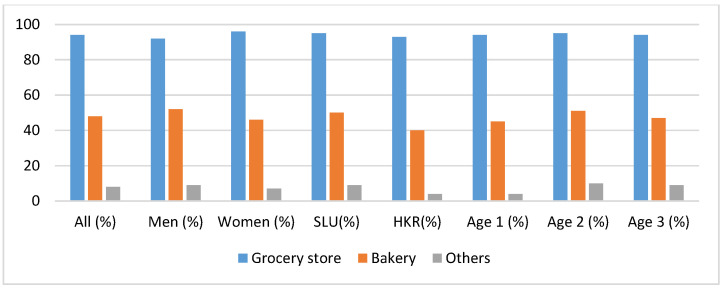
Preference of purchasing sites for heritage cereal products in the future. The category “Others” included the following options: Delivery to home/to work, REKO-ring, on-line/online store, directly from producer/grower, café, farm shop, bake yourself and market.

**Figure 7 foods-09-00742-f007:**
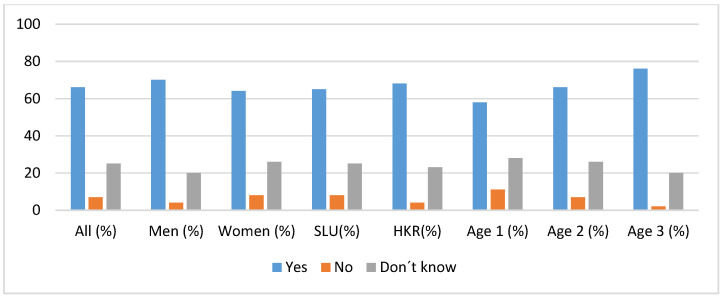
Frequency in percentages showing willingness to pay more for heritage cereal products.

**Table 1 foods-09-00742-t001:** Areas and questions covered in the web-based questionnaire.

Investigation Area	Indicators Used
A. Consumers’ awareness and consumption of heritage cereals	Approaches and habits of consuming bread and cereal based products. Awareness about different varieties of heritage cereals. Types and popularity of bread consumed. Accessibility, baking at home vs. purchasing site preference.
B. Consumers’ attitudes towards heritage cereals	Bread attributes that manipulate the choice of the bread. Receptiveness for new sorts of bread and cereal products that are based on heritage cereals. Willingness to pay more for bread and cereal products that are based on heritage cereals. Main attributes that would influence the choice of bread and products that are based on heritage cereals.
C. Consumers’ preferences of future products with heritage cereals	Kind of heritage cereals that are most likely to be consumed. Preference to accessibility: baking at home vs. purchasing sites.

**Table 2 foods-09-00742-t002:** Demographical distribution of the study population. Age in years.

Population Sector	Mean	Standard Deviation	Range
All	40.3	±15.0	19–91
Women	39.1	±15.0	19–91
Men	43.1	±14.8	21–74
SLU participants	40.1	±14.8	20–74
HKR participants	40.8	±15.7	19–91
Age Group 1	24.5	±2.6	19–30
Age Group 2	40.4	±5.6	31–50
Age Group 3	59.5	±6.8	51–91

**Table 3 foods-09-00742-t003:** Mean and standard deviations showing the importance of essential characteristics of the bread. Significant differences are indicated with different letters. A 5-pointed scale was used where five was regarded as a very important factor, while one represented the least important factor.

Character	All M ± std	Men M ± std	Women M ± std	X M ± std	Age 1 M ± std	Age 2 M ± std	Age 3 M ± std	HKR M ± std	SLU M ± std
Taste/Flavour	4.7 ± 0.6	4.7 ± 0.5	4.7 ± 0.1	4.6 ± 0.9	4.7 ± 0.6	4.8 ± 0.6	4.8 ± 0.5	4.7 ± 0.6	4.7 ± 0.5
Freshness	4.3 ± 0.8	4.2 ± 0.8	4.4 ± 0.9	4.8 ± 0.4	4.1 ± 0.9 ^a^	4.5 ± 0.7 ^b^	4.5 ± 0.7 ^b^	4.5 ± 0.8 ^a^	4.3 ± 0.9 ^b^
Texture	4.2 ± 0.8	4.0 ± 0.9 ^a^	4.3 ± 0.8 ^b^	4.2 ± 0.8 ^ab^	4.1 ± 0.9	4.3 ± 0.7	4.2 ± 0.9	4.3 ± 0.8	4.2 ± 0.8
Origin	3.8 ± 1.2	3.4 ± 1.3 ^a^	4.0 ± 1.1 ^b^	3.4 ± 1.8 ^ab^	3.8 ± 1.2	3.8 ± 1.2	3.9 ± 1.2	3.7 ± 1.3	3.9 ± 1.1
Odour/Aroma	3.7 ± 1.1	3.6 ± 1.0	3.7 ± 1.1	3.8 ± 1.3	3.3 ± 1.1 ^a^	3.9 ± 0.9 ^b^	3.9 ± 0.9 ^b^	3.7 ± 1.0	3.7 ± 1.1
Health	3.6 ± 1.1	3.4 ± 1.1	3.7 ± 1.0	3.4 ± 1.9	3.3 ± 1.1 ^a^	3.6 ± 1.8 ^b^	3.9 ± 1.0 ^c^	3.7 ± 1.1	3.6 ± 1.1
Wholemeal	3.5 ± 1.1	3.2 ± 1.2 ^a^	3.7 ± 1.0 ^b^	3.6 ± 1.7 ^ab^	3.7 ± 1.2 ^a^	3.6 ± 1.0 ^b^	3.7 ± 1.0 ^a^	3.7 ± 1.1	3.5 ± 1.1
Shelf life	3.3 ± 1.1	3.2 ± 0.9	3.3 ± 1.2	3.6 ± 1.7	3.3 ± 1.1	3.4 ± 1.1	3.3 ± 1.1	3.4 ± 1.1	3.3 ± 1.1
Sourdough	3.1 ± 1.2	3.1 ± 1.2	3.1 ± 1.2	3.0 ± 1.4	2.9 ± 1.3 ^a^	3.1 ± 1.2 ^ab^	3.4 ± 1.1 ^b^	3.4 ± 1.3 ^a^	3.0 ± 1.1 ^b^
Appearance	3.1 ± 1.1	3.0 ± 1.1	3.2 ± 1.1	3.0 ± 1.0	3.0 ± 1.1	3.2 ± 0.9	3.2 ± 1.1	3.3 ± 1.0 ^a^	3.1 ± 1.1 ^b^
Price	3.1 ± 1.0	3.0 ± 0.9	3.1 ± 1.0	2.8 ± 2.8	3.4 ± 1.0 ^a^	3.0 ± 1.0 ^b^	2.7 ± 0.9 ^c^	3.1 ± 1.1	3.0 ± 1.0
Organic	2.8 ± 1.3	2.6 ± 1.4	2.9 ± 1.3	3.8 ± 0.8	2.4 ± 1.2 ^a^	3.1 ± 1.3 ^b^	3.0 ± 1.3 ^b^	2.9 ± 1.3	2.8 ± 1.3
Brand	2.1 ± 1.1	2.0 ± 1.1	2.1 ± 1.1	1.2 ± 0.4	2.2 ± 1.1	3.0 ± 1.1	3.0 ± 1.1	2.1 ± 1.1	2.1 ± 1.1

Significant differences are indicated with different letters. SLU: Swedish University of Agricultural Sciences; HKR: Kristianstad University, Sweden; M: mean; std: standard deviation.

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
