# Peer review of "Consumer Awareness, Attitudes and Preferences Towards Heritage Cereals"

_foods, 2020, doi:10.3390/foods9060742_

Round 1
Reviewer 1 Report
The main question addressed by the research is whether heritage cereals consumers differ in academic and geographic backgrounds while taking age and gender into account. I consider that the topic is relevant and interesting. The topic is original, there are few previous studies which analyze this topic, essentially related to consumer awareness attitudes and interest towards heritage cereals. And the paper is well written, the text is clear and concise.
Some minor questions:
In the introduction section, you should try to find previous paper which analyzed the consumer perception towards Heritage Cereals, from different countries. the results should be commented by referring to these studies.
Raws 104-106 should be deleted.
Author Response
List of changes and answers to reviewer 1:
Comments and Suggestions for Authors, Review 1
The main question addressed by the research is whether heritage cereals consumers differ in academic and geographic backgrounds while taking age and gender into account. I consider that the topic is relevant and interesting. The topic is original, there are few previous studies which analyze this topic, essentially related to consumer awareness attitudes and interest towards heritage cereals. And the paper is well written, the text is clear and concise.
Thank you very much for this comment!
Some minor questions:
In the introduction section, you should try to find previous paper which analyzed the consumer perception towards Heritage Cereals, from different countries. the results should be commented by referring to these studies.
We agree that this would improve the text, we have not found consumer data for a specific country, but has added information on specific cereal: Rows 34-35. A new sentence is added: High acceptability has for example been shown for breads containing Kamut or spelt.
In lines 245 the following is added: ….and it has been shown that the acceptance of spelt is high among consumers.
Raws 104-106 should be deleted.
Rows 104-106 are deleted (now lines: 111-113)

Reviewer 2 Report
The paper is aimed at investigating and analyzing consumers’ awareness, attitudes and preferences towards heritage cereals. The topic is worth of interest and the writing style very clear. However the paper presents various limitations, starting from the statistical analyses performed to the presentation of the results. Moreover some points may need further explanation and the results obtained are poorly discussed or sometimes not discussed in a proper manner.
Below detailed major point.
Introduction:
Lines 63-71: Since the aim of the paper is to investigate and analyze consumers’ awareness, attitudes and preferences towards heritage cereals, this reviewer suggests to slightly improve this paragraph which seems to be limited in extension in respect to the previous part on heritage cereals.
Materials and methods
Questionnaire
Lines 87-92: Please insert the questions and the possible answers reported in the questionnaire (or at least report them as supplementary materials)
Data analysis: comments below
Results
Lines 104-106: to be deleted!
Participants: please merge table 2a and b and report the values regarding # of participants, mean, std deviation and range of SLU and HKR participants in the text
Lines 119-124: please clarify if these results are just a percentage of answers or some kind of statistical analyses have been run. If the author declare that ‘a variety of heritage cereal is know significantly more than another’, F and p values should be reported. However, the authors should be aware about the number of subjects in each group (e.g. X group with 6 individuals in respect to female group with more than 300 individuals).
Moreover, if some kind of analysis has been performed, please detailed better in the data analysis (e.g. multifactor anova or one way anovas or different analyses, fixed factors, dependent variables..etc)
Fig 1 – 2. Results presented in this way could be confusing. Please substitute Figures 1 and 2 with tables.
Lines 134-140: even if this reviewer appreciate this kind of results representation, the authors should explain: to which question is related? Probably to point c of area of investigations…please clarify and give much more details.
Lines 146-148: move to the materials and methods
Paragraph 4.3 and 4.4: these analyses should be run again! Below some general comments regarding these paragraphs.
No F and p values are presented, graphs not clear, no std dev or std errors are reported above the bars.. any significant results reported with letter or asterisks…
Please if multiple anovas have been run, consider to exclude 6 subjects in the unidentified gender. This reviewer completely understands the importance of gender issue but this is not the topic of the paper and the number of gender X (n=6) is not relevant compared to number of women and men. Please exclude these individuals, at least for the analytical analyses. Moreover, the analyses should take into account the unbalanced sample size in women and men groups (this limitation should be reported also in the discussion part).
It is still not clear in the results the importance of splitting the participants in respect of the Universities. This reviewer understand the possible influence of academic field of participants, but no differences seem to characterised the two different ‘populations’ in their attitudes or preferences. Moreover, any information about their educational level could be provided, even if all of them are supposed to be characterized by high education level. Take into account this consideration when the authors have to report the results in graphs.
As suggestion you could consider to report in graphs representation only the variables that showed a significant result. Moreover, different graphs could be prepared to separate the socio-demographics variables considered (e.g. age and gender separately), avoiding to put all together the results.
Lines 157 – 159: move to materials and methods
Discussion
Lines 225-227: Is it relevant? A non-Swedish reader does not understand the meaning of this phrase.
General comment: please improve the whole discussion.
The differences in age are the drivers of attitudes and preferences, but seem to be slightly deepened and the relationships highlighted in the results session discussed in a confounding manner or just mentioned.
To note the clearest and well discussed part is in lines 268-276 but probably is better to moved it to conclusion.
Gender differences are not discussed anywhere, just mentioned in the conclusion paragraph…
Other papers on this topic or related ones?
Lines 279-281: still uncertain about the scientific conclusion behind this comparison. Is it relevant for sustainability of agricultural system or heritage cereals production these differences on muesli or corn flakes consumption between the two areas? If yes, please clarify.
Author Response
List of changes and answers to reviewer 2:
Comments and Suggestions for Authors, Review 2.
The paper is aimed at investigating and analyzing consumers’ awareness, attitudes and preferences towards heritage cereals. The topic is worth of interest and the writing style very clear. However the paper presents various limitations, starting from the statistical analyses performed to the presentation of the results. Moreover some points may need further explanation and the results obtained are poorly discussed or sometimes not discussed in a proper manner.
Thank you for supporting this manuscript.
Below detailed major point.
Introduction:
Lines 63-71: Since the aim of the paper is to investigate and analyze consumers’ awareness, attitudes and preferences towards heritage cereals, this reviewer suggests to slightly improve this paragraph which seems to be limited in extension in respect to the previous part on heritage cereals.
The sentence now in line 67-68 is slightly changed and now reads: Consumers’ attitudes and preferences for different kind of products may differ according to factors such age, gender, education level and geographic background.
Materials and methods
Questionnaire
Lines 87-92: Please insert the questions and the possible answers reported in the questionnaire (or at least report them as supplementary materials)
A supplement (Supplement 1) with questions and possible answers is added, indicated in line 98-99.
Data analysis: comments below
Results
Lines 104-106: to be deleted!
Rows 104-106 are deleted (now lines: 111-113)
Participants: please merge table 2a and b and report the values regarding # of participants, mean, std deviation and range of SLU and HKR participants in the text
Table 2b is deleted and the content is included in the text line 118-120.
Lines 119-124: please clarify if these results are just a percentage of answers or some kind of statistical analyses have been run. If the author declare that ‘a variety of heritage cereal is know significantly more than another’, F and p values should be reported. However, the authors should be aware about the number of subjects in each group (e.g. X group with 6 individuals in respect to female group with more than 300 individuals).
Moreover, if some kind of analysis has been performed, please detailed better in the data analysis (e.g. multifactor anova or one way anovas or different analyses, fixed factors, dependent variables..etc)
In the section Statistical evaluation (line 101-109) it is stated that significant level is set to p<0.05. In lines 131, 147, 213, 226 and 234. is added that it is Chi2-analysis that have been applied on the frequency data. In line 162 and 183 it is added that one-way ANOVA has been applied. F- and p-values may be given for ANOVA-analyses, but not for Chi-squared tests.
Fig 1 – 2. Results presented in this way could be confusing. Please substitute Figures 1 and 2 with tables.’’
In our opinion it is more illustrative to show results as diagrams than in tables. In lines 128-129 it is added frequencies. Also captions in fig 1 and 2 are clarified.
Lines 134-140: even if this reviewer appreciate this kind of results representation, the authors should explain: to which question is related? Probably to point c of area of investigations…please clarify and give much more details.
Information of the question asked is now added in line 146.
Lines 146-148: move to the materials and methods
The lines are removed
Paragraph 4.3 and 4.4: these analyses should be run again! Below some general comments regarding these paragraphs.
No F and p values are presented, graphs not clear, no std dev or std errors are reported above the bars.. any significant results reported with letter or asterisks…
Tables with F and p values are now added. Tables 3 and 4
Please if multiple anovas have been run, consider to exclude 6 subjects in the unidentified gender. This reviewer completely understands the importance of gender issue but this is not the topic of the paper and the number of gender X (n=6) is not relevant compared to number of women and men. Please exclude these individuals, at least for the analytical analyses. Moreover, the analyses should take into account the unbalanced sample size in women and men groups (this limitation should be reported also in the discussion part).
Significant differences are clearly reported in the text, lines 184-203 To make the results clearer tables with mean and standard deviations are added as supplements 2 and 3, indications of significant differences are included.
Limitations on sample size are added to the discussion, lines 328-334.
It is still not clear in the results the importance of splitting the participants in respect of the Universities. This reviewer understand the possible influence of academic field of participants, but no differences seem to characterised the two different ‘populations’ in their attitudes or preferences. Moreover, any information about their educational level could be provided, even if all of them are supposed to be characterized by high education level. Take into account this consideration when the authors have to report the results in graphs.
Differences due to universities are given in the text. Focus for the different universities are given in material and method-section. However, level of education was not included in the study.
As suggestion you could consider to report in graphs representation only the variables that showed a significant result. Moreover, different graphs could be prepared to separate the socio-demographics variables considered (e.g. age and gender separately), avoiding to put all together the results.
Presenting in different graphs is a good idea but would yield very many diagrams and the overview will be lost. Instead the different groups as well as indications of significant differences are given in supplements 2 and 3.
Lines 157 – 159: move to materials and methods
The main part of the text is removed. One rewritten sentence introduce the reader to the results, now line 176-180
Discussion
Lines 225-227: Is it relevant? A non-Swedish reader does not understand the meaning of this phrase.
It may explain differences between Swedish and other European consumers and is therefore of relevance.
General comment: please improve the whole discussion.
The differences in age are the drivers of attitudes and preferences, but seem to be slightly deepened and the relationships highlighted in the results session discussed in a confounding manner or just mentioned.
To note the clearest and well discussed part is in lines 268-276 but probably is better to moved it to conclusion.
Gender differences are not discussed anywhere, just mentioned in the conclusion paragraph…
Discussions on age and gender are added in lines 293-316
Other papers on this topic or related ones?
The discussion is updated with more references, lines 298-334.
Lines 279-281: still uncertain about the scientific conclusion behind this comparison. Is it relevant for sustainability of agricultural system or heritage cereals production these differences on muesli or corn flakes consumption between the two areas? If yes, please clarify.
The statement has been clarified with a possible explanation, lines324-326.
Clarification of the sentence, line 277.

Round 2
Reviewer 2 Report
The quality of the paper has been improved but not sufficiently.
Graphs are still unclear and confusing for the reader, and few suggestions have been taken into consideration by the authors. Below detailed major point.
Introduction:
Lines 76-83: The purpose of the previous comment was not to slightly change the phrase, which was already clear as it stood, but to add more comments on this paragraph. Please improve.
Results
Lines 173-174. The authors stated: ‘Chi squared tests showed that this variety was significantly more known to HKR than to SLU’. As previously required please add the statistical values. The phrase should be: ‘Chi squared tests showed that this variety was significantly more (χ2= …; p < 0.05) known to HKR than to SLU’. Do the same in the following lines. Fig 1 – 2. In this reviewer opinion these results are still not clear. Please, at least, divide the figure in 3 different graphs regarding age, gender and institutions.
Moreover, again, is statistically incorrect to compare 6 subjects against 311 women and 117 men, using a simple ANOVA or Chi-squared test. Please do not include the group ‘X’ in the analyses.
The limitations added are not sufficient to justify the inclusion for the analytical analyses.
Paragraph 4.3 and 4.4 These two paragraph still present limitations:
- It is not necessary report the F and p values in table 2. It is sufficient to add in the text in the proper manner (e.g. lines 226-227: However, it was significantly more common (F=11.30; p<0.001) to purchase bread at bakery shops in age groups 2 and 3 compared to age group 1. Please delete table 2 and report the F and p-values in the text, where needed!
- Bars should present the standard error or standard deviation, it could not be presented in the supplementary files only.
- Please, again, consider to show only the relevant results (e.g. in figure 4 represents the age groups, which is the only significant result!)
- Figure 5 could be substituted with table in supplement 3 adding the F and p values reported in table 4 of the main text
- In Figure 6 could consider only age and gender groups.
- Delete figure 7 since no significant results have been highlighted from the analyses
Discussion
Lines 428-432: Are the authors stating that the multidisciplinary focus of the HKR influenced the major consumption of muesli or corn flakes? Please clarify.
Author Response
List of changes and answers to reviewer:
Comments and Suggestions for Authors
The quality of the paper has been improved but not sufficiently.
Thank you for reviewing this manuscript again.
Graphs are still unclear and confusing for the reader, and few suggestions have been taken into consideration by the authors. Below detailed major point.
All figures are changed and now show each consumer group, gender group X is excluded. Also in Table 2, gender group X is excluded
Introduction:
Lines 76-83: The purpose of the previous comment was not to slightly change the phrase, which was already clear as it stood, but to add more comments on this paragraph. Please improve.
A new sentence giving more explanation regarding consumption of bread and a reference on Swedish bread consumption is added.
Results
Lines 173-174. The authors stated: ‘Chi squared tests showed that this variety was significantly more known to HKR than to SLU’. As previously required please add the statistical values. The phrase should be: ‘Chi squared tests showed that this variety was significantly more (χ2= …; p < 0.05) known to HKR than to SLU’. Do the same in the following lines. Fig 1 – 2. In this reviewer opinion these results are still not clear. Please, at least, divide the figure in 3 different graphs regarding age, gender and institutions.
χ2- and p-values are given in the text, lines 159-160, 231-236 and 258. All figures are changed and now show each consumer group, gender group X is excluded.
Moreover, again, is statistically incorrect to compare 6 subjects against 311 women and 117 men, using a simple ANOVA or Chi-squared test. Please do not include the group ‘X’ in the analyses.
Gender group X is excluded, this is indicated in lines 121-123.Group X is not included in the analysis, this is indicated in the limitation paragraph, line 355.
The limitations added are not sufficient to justify the inclusion for the analytical analyses.
Group X is not included in the analysis, this is indicated in the limitation paragraph, line 355.
Paragraph 4.3 and 4.4 These two paragraph still present limitations:
- It is not necessary report the F and p values in table 2. It is sufficient to add in the text in the proper manner (e.g. lines 226-227: However, it was significantly more common (F=11.30; p<0.001) to purchase bread at bakery shops in age groups 2 and 3 compared to age group 1. Please delete table 2 and report the F and p-values in the text, where needed!
F- and p-values are reported in the text, lines 178, 201-202, 209-211, 213-214, 216-218. The tables reporting F- and p-values are deleted.
- Bars should present the standard error or standard deviation, it could not be presented in the supplementary files only.
It is of our opnion that bars will make the figures messy and hard to read, therefore mean and standard deviations are added as tables (tables 3 and 4), indications of significant differences are given.
- Please, again, consider to show only the relevant results (e.g. in figure 4 represents the age groups, which is the only significant result!)
It is also of interest to show the differences between where to buy or buy at home. Therefore we would like to keep the figure. We suggest that editor may decide.
- Figure 5 could be substituted with table in supplement 3 adding the F and p values reported in table 4 of the main text.
Tables 4 is added, however we keep the figure as illustration and let the editor decide if it should be deleted.
- In Figure 6 could consider only age and gender groups.
In order to make it easy for the reader it is our aim to present all figures in the same way, where all consumers groups are shown,
- Delete figure 7 since no significant results have been highlighted from the analyses
In parallel with keeping figure 4, it may be of interest to the reader to see not only differences between consumer groups, but also differences in where/how the consumers would like to get their bread. Also here the editor may decide how to do.
Discussion
Lines 428-432: Are the authors stating that the multidisciplinary focus of the HKR influenced the major consumption of muesli or corn flakes? Please clarify.
The sentence is now rewritten and includes all differences between the sites:Lines 350-352
